# Abscisic Acid, Microtubules and Phospholipase D-Solving a Cellular Bermuda Triangle

**DOI:** 10.3390/ijms26010278

**Published:** 2024-12-31

**Authors:** Xuan Liu, Michael Riemann, Peter Nick

**Affiliations:** 1State Key Laboratory of Crop Stress Biology for Arid Areas (Shaanxi Key Laboratory of Apple), College of Horticulture, Northwest A&F University, Yangling 712100, China; lx156273@163.com; 2Molecular Cell Biology, Joseph Gottlieb Kölreuter Institute for Plant Sciences, Karlsruhe Institute of Technology, Fritz-Haber-Weg 4, 76131 Karlsruhe, Germany; michael.riemann@kit.edu

**Keywords:** abscisic acid, microtubules, tubulin tyrosine ligase, tubulin tyrosine carboxypeptidase, detyrosination, tyrosination, Phospholipase D, cold stress

## Abstract

Rice plants are important food crops that are sensitive to cold stress. Microtubules (MTs) are highly associated with plant response to cold stress. The exogenous application of abscisic acid (ABA) can transiently induce the cold stability of microtubules. These phenotypes were accompanied by the transient increase in Phospholipase D (PLD) enzyme activity. The analysis of detyrosinated/tyrosinated α-tubulin by Western blot in the NtTUA3 line or in the NtTUA3+OsTTL line gave us such a conclusion that the effect of ABA on detyrosinated α-tubulin not only was regulated by ABA but also was dependent on the TTLL12 protein. The dual ABA and 1% n-butanol treatments had shown that ABA-induced detyrosinated α-tubulin in a manner distinct from the n-butanol pathway. Detecting the detyrosinated α-tubulin level after pre-treatment with pertussis toxin (PTX), a G-protein inhibitor, followed by ABA, as well as mastoparan (Mas7) treatment suggested that the effect of ABA on detyrosinated α-tubulin was dependent on PLD activity.

## 1. Introduction

Climate change is usually discussed in the context of heat and drought during summers. However, it can also mean untimely cold stress. Unusually warm temperatures during early spring provoke the precocious development of plants such that they become sensitive to subsequent cold snaps with devastating consequences. The ability to sense, process, and adapt to cold stress rapidly and efficiently shifts into the focus, therefore. While the transcriptional cascade regulating the expression of cold-responsive (so-called COR) genes has been intensively addressed by molecular genetics, mostly in thale cress as a model [1], upstream signalling has not attained the same degree of attention [2].

In this context, microtubules are crucial. While their role in cell shape and internal structure is well known, it becomes progressively clear that they also act in the perception and processing of abiotic stress signals [2,3]. Cold stress is a physical signal and needs to be converted into a chemical signal to launch transduction. Microtubules have been proposed as susceptors able to collect and focus the minute mechanical forces caused by the rigidifying membrane to breach the threshold of thermal noise and generate an input that culminates in the activation of calcium influx, leading to a signalling cascade involving membrane-located NADPH oxidases, but also G-protein-dependent Phospholipase D signalling. The signalling deployed by these susceptive and perceptive events targets also microtubules themselves, thus creating a self-amplifying loop that will generate a clear input for downstream signalling towards the nucleus [2]. As to be expected from such a sensory role, the transient elimination of microtubules is necessary and sufficient to induce subsequent freezing tolerance [4,5].

However, microtubules are not only acting upstream in sensory transduction, but they are also modified as a result of cold stress. For instance, they can acquire freezing resistance after prolonged chilling in the context of so-called cold acclimation, often following a transient elimination during early phases of chilling [2,6]. The cold stability of microtubules can also be induced by abscisic acid [7]. Moreover, abscisic acid is a well-known inducer of cold hardiness [8], leading to the concept that cold acclimation is mediated by the induction of abscisic acid [9]. However, for microtubules, a comparative study in winter wheat [10] showed that while both treatments can induce their cold stability, the response of microtubules to abscisic acid and cold acclimation differs qualitatively. Thus, the effect of cold acclimation on microtubules seems to involve a second pathway that is independent of abscisic acid. A similar conclusion has also been drawn from a comprehensive review of gene regulation in response to either cold acclimation or abscisic acid [11].

The cold stability of microtubules, either in response to cold acclimation or in response to ABA, is probably linked with a changed interaction between microtubules and plasma membrane. In fact, classic work [12] has demonstrated that proteins in the cell wall are able to stabilise cortical microtubules against cold stress. This phenomenon has to be seen in the context of the so-called cytoskeleton–cell wall continuum, coined in analogy to the integrins found in animal cells [13]. A biochemical approach searching for such proteins that link microtubules with the plasma membrane led to the identification of Phospholipase D [14]. Since Phospholipase D (PLD) forms a covalent bond with the phospholipids in the membrane, before it transfers the acyl moiety to water, it should mediate a transient binding of microtubules with the plasma membrane [15]. Depending on the activity of PLD and the lifetime of this intermediate state, there should be a steady-state fraction of microtubules linked to the membrane. The activation of PLD by signals should reduce the duration of this intermediate state, expected to become manifest as a net detachment of microtubules from the membrane. Recently, the effect of n-butanol as a primary alcohol on PLD activity has been reported. N-butanol is a specific inhibitor of PLD signalling. In the presence of n-butanol, it can interfere with the accumulation of phosphatidic acids, products of PLD with signal activity, and eliminate cortical microtubules [16].

Several lines of evidence support a link between microtubules, PLD, and cold stress mediated by the activation of the plant trimeric G-protein. A specific PLD, PLDα1, has been shown to interact with the plant G-protein through a specific protein motif [17]. In fact, the modulation of G-protein activity by mastoparan-induced microtubule detachment [18]. Conversely, the cold-induced disassembly of microtubules was efficiently blocked by pertussis toxin, a specific inhibitor of G-proteins [19]. The same PLD type is also required for ABA responses. The suppression of PLDα1 by an antisense strategy interfered with the ABA response of senescence in the leaves of Arabidopsis [20]. The trimeric G-protein seems also involved in cold tolerance–overexpression of the α-subunit rendered tomato, a chilling-sensitive species, cold-tolerant [21], and the chilling tolerance of japonica rice is mediated by a specific allele of COLD1, a regulator of G-protein activity [22].

The cold stability of microtubules through ABA or through cold acclimation might be achieved either by interfering with the activation of PLD in response to cold or by altering the interaction of microtubules with PLD. Our previous work has already collated evidence for the second mechanism (which per se does not exclude the possibility that the first mechanism is at work as well).

## 2. Results

### 2.1. ABA Can Increase the Cold Stability of Cortical Microtubules

The induction of cellular resistance to cold stress is often linked with an accumulation of abscisic acid (ABA), often accompanied by an elevated cold tolerance of microtubules [10]. We, therefore, addressed the role of ABA in the cold response of microtubules in tobacco BY-2 cells, expressing the tubulin marker TuA3-GFP. In the absence of cold stress, we did not detect any significant effect of 50 µM ABA (administered for 60 min) compared to the control (Figure 1A,B). Likewise, treatment for 60 min with 500 µM of AA1, an inhibitor of ABA signalling [23], did not cause any alteration of the microtubules (Figure 1C). In contrast, the cold response of the microtubules was modulated when the ABA status was manipulated. When control cells were subjected to 0 °C for 60 min, the transverse arrays of microtubule bundles seen if cells were kept at 27 °C mostly vanished, with individual and thinner microtubules remaining in the background of a cytosolic tubulin-GFP signal (Figure 1D). This microtubule elimination was clearly mitigated by a pre-treatment with 50 µM ABA for 60 min (Figure 1E). In contrast, a pre-treatment with 500 µM AA1 accentuated the cold-induced elimination of microtubules (Figure 1F). To test the significance of these differences, we quantified microtubule integrity under these conditions (Figure 2). We observed that a cold treatment of 60 min 0 °C reduced microtubule integrity by around 40% (Figure 2A). Pre-treatment by ABA reduced this loss in integrity in response to a subsequent cold treatment to 23%. This is to be seen also in relation to the fact that ABA alone did not alter microtubule integrity. When cells were treated with the ABA inhibitor AA1, this did not affect microtubule integrity; there was even a mild, albeit not significant, increase of around 10% (Figure 2B). However, the subsequent response to the cold treatment was accentuated significantly (55% decrease in integrity) over the response seen for cold stress alone (40% decrease in integrity). Thus, ABA causes the stabilisation of microtubules against cold stress, and the disruption of ABA signalling enhances the elimination of microtubules by cold. When we tested whether prolonged ABA pre-treatment might promote this stabilising effect (Appendix A), we found that a pre-treatment for 120 min or for 180 min was less effective, although still leading to a slight stabilisation as compared to cold stress without ABA pre-treatment.

### 2.2. ABA and Cold Can Transiently Increase the Activity of Phospholipase D

Phospholipase D (PLD) has been described as a hub for stress signalling, including for low temperatures, and has been identified as a necessary component for the microtubular response to cold [22]. We therefore measured the enzymatic activity of PLD using a colorimetric assay [24] in response to ABA (60 min, 50 µM) and in response to cold stress (60 min, 0 °C). Both factors increased PLD activity significantly—by almost 50%—over the resting level (Figure 3A). The combination of ABA and cold stress did not yield an additive effect, the resulting stimulation of PLD activity was no different from that seen for each factor alone. To assess whether the stimulation in response to cold stress was dependent on ABA, we blocked ABA signalling by AA1 (Figure 3B). While AA1 administered in the absence of cold stress did not cause any change in PLD activity, it did not interfere with the stimulation caused by cold stress. Thus, while ABA can stimulate PLD activity, ABA signalling seems to be dispensable for the stimulation of this enzyme by cold. To understand the temporal dynamics of enzyme stimulation, we measured PLD activity over the increasing duration of the ABA treatment (Appendix A) and found that this effect was transient. At 2 h after the addition of ABA, the stimulation had already dropped to around 25% above the resting level, and after 3 h, it had already fully dissipated.

### 2.3. ABA Can Transiently Increase the Pool of Detyrosinated Tubulin

Since pre-treatment with ABA can stabilise microtubules against subsequent cold stress (Figure 1A and Figure 2A), we wondered whether this stabilisation might be linked with a reduced turnover of microtubules. Since the lifetime of microtubules correlates with post-translational modifications such as the detyrosination of α-tubulin, we addressed the ABA response of the tyrosinated versus the detyrosinated tubulin pools using monoclonal antibodies to differentiate between the two forms of α-tubulin (Figure 4). The antibody raised against tyrosinated α-tubulin was recognised through a characteristic double band (Figure 4A) at around 55 kDa, whereby the upper band corresponds to the tyrosinated form, while the lower band is caused by a cross-reaction with the detyrosinated form that is sometimes but not reliably observed. In addition, a higher band of above 80 kDa was detected that corresponds to the fusion of tobacco α3 tubulin and GFP present in this marker line [22]. While the band presumably reporting tyrosinated tubulin was persistent in response to ABA, the band stemming from the cross-reaction with detyrosinated tubulin significantly faded out, especially for prolonged treatment with ABA. Since the detection of this band comes from a cross-reaction and, therefore, cannot be used as a reliable readout for detyrosinated tubulin, we probed the same samples with a monoclonal antibody specially targeted to detyrosinated α-tubulin. This antibody did not show any cross-reaction with tyrosinated tubulin, such that here, only two bands were visible (the lower band of the tubulin duplex, and the band at around 80 kDa originating from the GFP fusion of tubulin). Here, the signal for detyrosinated tubulin first increased significantly at 1 h after the addition of ABA but subsequently decreased and at 3 h, had almost vanished. To quantify this effect, control and ABA-treated extracts were run side by side in numerous replications, such that the bands could be quantified relative to each other using the respective control (incubation in the absence of ABA) as a reference. This quantification (Figure 4B) showed that the pool of tyrosinated tubulin was at first not very responsive but remained comparable to the control over 2 h. However, after 3 h of ABA treatment, it dropped significantly, by around 40%. In contrast, the detyrosinated pool was strongly responsive with a transient induction of 60% after 1 h of ABA treatment and a subsequent decline to around 80% below the control level when the ABA treatment was prolonged to 3 h.

Thus, ABA can induce a transient stabilisation of microtubules against cold (Appendix A), a transient stimulation of PLD activity (Appendix A), and a transient increase in the detyrosinated pool of α-tubulin (Figure 4B). All three phenomena evoked by ABA show their peak at 60 min and subsequently disappear.

### 2.4. ABA Interaction with TTLL12 Protein Regulates Tubulin Modification in the Early Stage

Our previous work [25] had identified that PLD activity was regulated by rice tubulin tyrosine ligase-like 12 (TTLL12). To verify this hypothesis, we followed the response of ABA on detyrosinated/tyrosinated α-tubulin in the double transgenic cell line TUA3+TTL12. After 1 h and 2 h of ABA treatments, either in the TUA3 line (without TTLL12) or the TUA3+TTLL line (with TTLL12), neither the bands around 50 kD reporting tyrosinated α-tubulin by using the monoclonal antibody ATT nor reporting detyrosinated α-tubulin by using the monoclonal antibody DM1A showed any significant differences. However, after 3 h of ABA treatment, the tyrosinated α-tubulin level dropped and increased significantly by around 30% and 20% in the TUA3 line and the TUA3+TTLL line, respectively (Figure 5). In contrast, compared to the transient increase after 1 h of ABA treatment in the TUA3 cell line, then with a subsequent decline to around 80% after 3 h of ABA treatment in the TUA3 cell line (Figure 5B), the detyrosinated α-tubulin level of the TUA3+TTLL cell line kept constant after 1 h and 2 h of ABA treatments, and it reduced by only around 20% below the control level after 3 h of ABA treatment (Figure 5D). However, in our expectation, the response of ABA on detyrosinated α-tubulin should be elevated after 1 h ABA treatment. We cannot forget that the system has to deal with a high level of TTLL already in the TUA3+TTLL cell line before ABA application, and induction would, therefore, not be visible. So, we definitely said that TTLL was in the loop.

### 2.5. The Response of Detyr-Tub to ABA Is Independent of the n-Butanol Pathway

Our previous results showed that the activation of PLD can produce PA, which can recruit the MAP651 protein to stabilise microtubules. And the OsTTLL12 can compete with MAP65-1 for the C-terminal binding site of α-tubulin, which can prevent the coupling of microtubules and MAP65-1, resulting in promoting the depolymerization of microtubules [22]. In order to test whether the effect of ABA on the detyrosinated α-tubulin level is dependent on the PA pathway, we used n-butanol with the property of blocking the PA signal to treat the TUA3 cell line.

After 1 h of double treatment with n-butanol and ABA, the abundance of detyrosinated α-tubulin was increased. Then, the abundance of detyrosinated α-tubulin was strongly inhibited and decreased by 97% after 2 h of double treatments with n-butanol and ABA (Figure 6D). However, double n-butanol and ABA treatment did not affect the tyrosinated α-tubulin level (Figure 6C). ABA treatment for 1 h induced transiently the detyrosinated α-tubulin level; then, it decreased gradually over 2 h and 3 h of ABA treatments (Figure 6B). The trend of tyrosinated α-tubulin level was the same as the double n-butanol and ABA treatment (Figure 6A). While the effect for detyr tubulin at 2 h looks impressive, it is just the sum of the response to n-butanol alone (seen after 2 h of addition) and the response to ABA (seen after 1 h of addition). The interaction of ABA and n-butanol on detyrosination is additive, which means, therefore, they go through different paths and only converge at their final point.

### 2.6. The Early Response of Detyr-Tub to ABA Is Dependent on PLD Activity

So far, ABA not only can activate PLD activity but also can stimulate detyrosinated α-tubulin. So, next, we wonder which part (PLD or detyrosinated α-tubulin) will be activated by ABA. Mas7, a G-protein activator, and PTX, an inhibitor of G-protein, were used to test this assumption. The tyrosinated α-tubulin and detyrosinated α-tubulin were induced after 1 h of Mas7 treatment. However, the regulation of tyrosinated α-tubulin was significantly stronger than that of the detyrosinated α-tubulin. In contrast, double treatment with PTX for 1 h and ABA for 1 h inhibited detyrosinated α-tubulin and tyrosinated α-tubulin levels by about 4% and 11%, respectively. If ABA first activated detyrosinated α-tubulin during blocking with PTX for 1 h, then followed by ABA for 1 h, the detyrosinated α-tubulin should have been induced. But now it was inhibited (Figure 7). Thus, from these results, we could say that ABA first stimulated PLD activity.

### 2.7. The Application of ABA Inhibited the Transcripts for NtTUA3-GFP

To understand why the NtTUA3 protein level was first strongly induced and then had a significant decline no matter whether it was in the ABA treatment or in the n-butanol treatment, we measured steady-state transcript levels of NtTUA3-GFP in the TUA3 cell line under the ABA and n-butanol conditions. We found that the NtTUA3-GFP transcript level did not change after ABA treatment. However, in the presence of n-butanol alone or ABA plus n-butanol together, the NtTUA3-GFP transcript level was reduced (Figure 8A). Therefore, these results indicated that ABA regulated NtTUA3 protein levels directly not through transcript level.

For the next step, to test the above hypothesis, we used MG132, a specific inhibitor of the proteasome, to treat the TUA3 cell line. We observed that the NtTUA3-GFP transcript level was induced strongly at 30 min; then, it was reduced gradually when cells were pretreated with 100 µM MG132 0.5 h, followed by 1 h, 2 h, and 3 h of ABA treatments. DMSO as a control solvent promoted NtTUA3-GFP transcript levels after a 30 min treatment, but then, it kept constant (Figure 8B). These results indicated that ABA regulated NtTUA3 protein level.

## 3. Discussion

### 3.1. A Role of ABA on the Cold Stability of Microtubules

The induction of cold resistance in microtubules will result in the accumulation of ABA content [10]. Now, the question is how the ABA accelerates the resistance of microtubules under cold stress. To obtain insight into the function of ABA in the cold resistance of microtubules, we used ABA in the present study to treat the overexpressed tubulin marker gene TUA3 in the tobacco BY-2 cells in the absence or presence of cold stress. We observed that ABA treatment for 60 min inhibited the disassembly of cortical microtubules under cold conditions (Figure 1A,B,D,E), which was matched with a report that the pre-treatment with ABA enhanced the cold resistance of microtubules [7,26,27]. In contrast, the ABA inhibitor AA1 accentuated the cold-induced elimination of microtubules (Figure 1A,C,D,F). This is consistent with the fact that ABA focuses on increasing microtubules’ stability against cold stress [28]. The ability of ABA to induce microtubular cold stability may be due to the increase in ABA levels induced by cold stress, which can result in the induction of specific proteins that are responsible for the increased cold stability [29].

### 3.2. ABA Interacts with PLD to Affect the Cold Stability of Microtubules

Phospholipase D (PLD), a 90 kDa protein that was isolated from tobacco BY-2 membranes, not only can decorate microtubules in plant cells but also has been demonstrated to act as a signal hub to link with the microtubular response to cold stress [30]. However, how does PLD function in the cold response of microtubules under ABA treatment? It is still not clear. We therefore probed for the effect of PLD by detecting the enzymatic activity of PLD under ABA conditions and cold stress conditions. We found that ABA for 60 min and cold for 60 min increased PLD activity significantly over the resting level (Figure 3A). Interestingly, we also found that after extending the treatment time to 180 min, the PLD activity had already fully dissipated (Appendix A). So, this effect was transient. For this situation, this effect is linked to the fact that PLD acts as an early step of cold signalling to increase plant tolerance [22]. It has been demonstrated that the phospholipase activity of PLD is dependent on the rapid increase at the intracellular level, which can be regulated by cold stress [31]. Furthermore, the PA, a second messenger, is produced by PLD, which can hydrolyse structural phospholipids, which can cause ABA-induced ion channels to close stomata in plants, resulting in the insensitivity of the cells of microtubules to ABA treatment [32,33].

### 3.3. ABA Results in Higher Abundance of Detyrosinated α-Tubulin by Affecting PLD Activity

So far, we have identified the role of ABA in the cold response of microtubules through mediating PLD enzyme activity. It was found that OsPLDα1 binds to detyrosinated α-tubulin and accelerates the production of more detyrosinated α-tubulin [16]. Now, there is a question: What is the role of detyrosinated α-tubulin in PLD-dependent ABA signalling? To test this relationship, we used monoclonal antibodies ATT and DM1A to detect the tyrosination of α-tubulin and detyrosination of α-tubulin, respectively. We found that ABA for 1 h first increased significantly the level of the detyrosination of α-tubulin but subsequently decreased and at 3 h, had almost disappeared (Figure 4). So, the effect of ABA on the detyrosination of α-tubulin was also transient. In the last step, we discussed how ABA could activate the PLD enzyme. And the OsPLDα 1 overexpressed in tobacco cells showed an increase in detyrosinated α-tubulin [22]. There is a new question: whether the increase in detyrosination α-tubulin levels is due to the activation of PLD enzyme by ABA directly. Since G-protein can activate PLD, we used PTX, an inhibitor of G-protein, to test whether there is a change in the detyrosination level of α-tubulin when PLD activity is blocked. We did not see a clear change in the detyrosination level of α-tubulin (Figure 7). Based on these results, we proposed that ABA stabilizes microtubules by mediating the PLD-dependent detyrosination pathway.

### 3.4. The Role of OsTTL in the Cold Stability of Microtubules Is Induced by ABA

Apart from the above results, our previous work [34] had identified that a tubulin tyrosine ligase-like 12 (OsTTLL12), which acts as the only candidate of TTL homologues in a plant, can bind to detyrosinated α-tubulin. Now, based on this conclusion, a new question appears. Does the TTLL12 protein participate in the whole loop? We used a suspension culture of tobacco (*Nicotiana tabacum* L. cv Bright Yellow-2, BY-2) cells co-expressing NtTUA3-GFP/OsTTLL12-RFP to test this hypothesis. ABA treatment did not alter the detyrosination/tyrosination of α-tubulin. These results were completely different from that in the ABA treatment of the NtTUA3-GFP cell line (Figure 5). For this situation, we should not forget that α-tubulin is already in the co-expressing NtTUA3-GFP/OsTTLL12-RFP cell line. The TTLL12 protein is linked to the detyrosinated α-tubulin and also interferes with PLD [22], which would explain why the application of ABA affects the cycle of detyrosination/tyrosination of α-tubulin, which is dependent on the TTLL12 protein. In brief, when ABA is used on the co-expressing NtTUA3-GFP/OsTTLL12-RFP cell line, according to our above results, it will induce a cascade reaction, in which ABA will stimulate PLD activity, resulting in increased levels of the detyrosination of α-tubulin. At the same, the TTLL12 protein also induced PLD activity and increased levels of the detyrosination of α-tubulin. We therefore think ABA should mitigate the TTLL12 protein.

### 3.5. The Relationship Between Rice TTL and TTC

N-butanol is a primary alcohol and has the property of activating PLD, which can produce phosphatidic acid (PA). Apart from the production of PA by n-butanol treatment, it can also consume PA. In brief, when n-butanol activates PLD, it is the same as n-butanol itself also acting as a trans-phosphatidylation substrate [22,34]. This is because of the specific function of n-butanol, which can consume PA, in addition to the fact that PA can also recruit MAP65-1. Therefore, when PA is consumed by n-butanol, it can result in lower levels of MAP65-1, which will compete with OsTTLL12 for the C-terminal binding sites of α-tubulin, leading to the increase in OsTTLL12 levels. Since more OsTTLL12 will prevent the inhibition of tyrosinated αβ tubulin dimers on the translation of α tubulin, detyrosinated α-tubulin will be induced [22]. In our work, a 55 kDa sized protein detected by Western blot using antibodies ATT and DM1A proved that the content of detyrosinated α-tubulin was induced significantly by n-butanol treatment alone at 2 h. The result we obtained was consistent with the fact that n-butanol can activate PLD, which binds to detyrosinated α-tubulin [22]. Therefore, we proposed that in the presence of n-butanol, detyrosinated α-tubulin was favoured such that more TTC, not TTL, was produced. There was a contradiction with the evidence that OsTTLL12, as the most likely plant homologue for TTL function, could lead to a higher level of detyrosinated α-tubulin [25]. We guessed whether TTL is a TTC. If this assumption is correct, the results we obtained seem to be reasonable.

Although TTL has been found in the porcine brain, TTC is still unclear [35]. The vasohibins (VASHs), as well as their regulator the Small Vasohibin-Binding Proteins (SVBPs), have been shown to act as the longtime elusive tubulin carboxypeptidases (TCPs) (in plant models, also abbreviated as TTC). But, so far, there are no homologues of these animal TCPs in plants [12,36]. A study has shown that the enzyme TTL prefers tyrosinated tubulin, and the enzyme TTC can yield detyrosinated tubulin [25]. If we assume that TTC is present in plants, when there exists more TTL, more tyrosinated tubulin will be produced. However, the overexpression of TTC will lead to more tyrosinated tubulin. Combined with our data, in the presence of ABA, the overexpression of OsTTLL12 decreased detyrosinated tubulin at 3 h, whereas the tyrosinated tubulin increased at 3 h (Figure 5C,D). So, we proposed that OsTTL might be also the TTC. Moreover, a TTC inhibitor named parthenolide has been found to decrease tubulin detyrosination contents in BY-2 tobacco cells [37], further suggesting TTL is a TTC.

However, we should keep in mind that the post-translational modification and tubulin protein synthesis may be involved in increasing tubulin detyrosination contents caused by OsTTL [38]. It has been shown that the complexes will be formed due to the combination of the LRR domain characteristics of OsTTLL12 [25] and other proteins. Because this LRR domain is different from the set-like domain of animals, it is conceivable that this interaction may guide the direction of the enzymatic reaction (connecting tyrosine vs. cutting tyrosine). In order to obtain the biochemical pathway of enzyme activity given by TTLL12 and potential binding partners, the recombinant expression will be carried out in future in vitro research.

### 3.6. The Effect of ABA on Detyrosinated α-Tubulin Is in a Manner Distinct from the n-Butanol Pathway

So far, we have addressed the fact that ABA will deploy the activation of G-protein, resulting in triggered PLD activation, which can interact with the TTLL12 protein to regulate the detyrosination/tyrosination cycle. However, phosphatidic acid (PA), a product of PLD activation, can couple with the MAP65-1 protein, which can compete with the TTLL12 protein for the C-terminal binding sites of α-tubulin to affect detyrosinated α-tubulin levels [22]. To investigate whether ABA and n-butanol have the same response path to detyrosinated α-tubulin, in our study, a 55 kDa sized protein detected by Western blot using antibodies ATT and DM1A, respectively, proved that the effect of ABA and n-butanol on detyrosination is additive. After 2 h, the effect of detyrosination is simply the sum of the response to n-butanol alone (seen after 2 h of addition) and the response to ABA (seen after 1 h of addition) (Figure 6), suggesting that the role of ABA in response to detyrosination is independent of the n-butanol pathway.

## 4. Materials and Methods

### 4.1. Cell Lines and Cultivation

A suspension culture of tobacco (*Nicotiana tabacum* L. cv Bright Yellow-2, BY-2) cells expressing the fluorescent tubulin marker GFP-NtTUA3 (termed TUA3) and a line expressing, in addition, the tubulin tyrosine ligase-like 12 from rice in fusion with RFP (termed TTL+TUA3) were used in this study [22]. These cell lines were cultivated in liquid Murashige–Skoog (MS) medium (4.3 g/L MS salts (Duchefa, Haarlem, The Netherlands)), supplemented with 30 g/L sucrose, 200 mg/L K_2_HPO_4_, 100 mg/L myo-inositol, 1 mg/L thiamine, and 0.2 mg/L 2,4-dichlorophenoxyacetic acid (2,4-D), pH 5.8. To sustain selective stringency, the respective antibiotics were added, 50 mg/L kanamycin in case of TUA3, or 30 mg/L hygromycin and 50 mg/L kanamycin in case of TTL+TUA3. Cells were sub-cultured by transfer of 1.5 mL cell suspension into 30 mL of fresh medium in a 100 mL Erlenmeyer flask. The subculture interval was biweekly. Cells were incubated in the dark at 27 °C under constant agitation on an orbital shaker at 150 rpm.

### 4.2. Treatment with Cold Stress and Abscisic Acid (ABA)

To address the function of microtubules in response to cold stress, TUA3 suspension cells collected on day 7 after sub-cultivation were transferred into 2 mL reaction tubes (Eppendorf, Hamburg), placed in a bath of ice water (0 °C), and incubated in the dark on a horizontal shaker. To monitor the effect of ABA on the cold response of microtubules, 2 mL of TUA3 cells were first pre-treated with 50 μM ABA (Duchefa, Harlem, The Netherlands) for defined time intervals (1 h, 2 h, and 3 h) at 25 °C before exposure to ice water for 1 h. Cells were observed by microscopy immediately afterwards using a pre-cooled slide and coverslip to delay re-warming. To address the role of endogenous ABA during the cold response, the inhibitor ANTAGONIST 1 (AA1, Life Chemicals, Niagara, ON, Canada) was administered at 500 µM at 25 °C for 1 h to TUA3 cells. This inhibitor blocks the binding pocket of the ABA receptor and, thus, disrupts, ABA signalling [23]. Subsequently, these cells were subjected to cold stress for 1 h before assessing the state of microtubule depolymerisation. During all treatments and incubations, the cells were kept in the dark while shaking at 150 rpm. Parallel to microscopical inspection, a second set of each sample was shock-frozen in liquid nitrogen to be used later for measuring the activity of Phospholipase D. In a second set of experiments, both TUA3 and TTL+TUA3 cells were treated with 50 µM ABA for defined time intervals (up to 5 h) at 25 °C in the dark to be shock-frozen and later used for protein analysis by Western blotting, probing for detyrosinated and tyrosinated α-tubulin.

### 4.3. Measuring Phospholipase D Activity

In plant cells, the activation of PLD is followed by the reorganisation of cortical microtubules [22]. Therefore, we measured PLD activity in vivo [24]. In brief, soluble proteins were extracted by spinning down cell debris with 1000× *g* for 15 min, transferring the supernatant to a fresh tube and centrifuging a second time with 15,000× *g* for 30 min to remove organelles [22]. The clear supernatant was then spun down with 105,000× *g* for 1 h to yield a sediment of microsomal membranes, which was then dissolved in 100 mM 3,3-dimethylglutaric acid (DMG) at pH 6.5 for the extraction of PLD. The enzyme activity was determined by a colorimetric method using 20 µL of the microsomal preparation in a total volume of 200 μL reaction mixture (10 mM MgCl_2_, 0.1 mM CaCl_2_, 100 mM DMG buffer (pH 6.5), 5 mM linoleic acid, and 12 mM phosphatidyl-choline). The reaction was incubated at 30 °C for 30 min and then directly transferred into boiling water for 15 min to denature the proteins. Subsequently, the reaction product was visualised by adding 800 µL of chromogenic solution (45 mM Tris-HCl, pH 8.0, 0.8 units choline oxidase, 2.4 units Horseradish Peroxidase (HRP), 0.24 mg oxidized 4-aminoantipyrine (4-ATT), and 0.16 mg phenol) incubated at 30 °C for 90 min. Prior to the readout, 1 mL of Tris-HCl (45 mM, pH 8.0) supplemented with 0.2 % *w*/*v* Triton X-100 was used to stop the reaction. To exclude any potential turbidity that might perturb the readout, the solution was filtered through a 0.22 μm syringe filter before measuring A_500_ [39]. Data represent mean and standard error from three biological replications.

### 4.4. Pharmacological Modulation of G-Protein-Dependent PLD Signalling

We addressed the role of G-protein activation and PLD activity for the response of TuA3 cells to ABA (50 µM) by pre-treatment with specific inhibitors or activators for 1 h, sampling the cells at 1 h, 2 h, and 3 h after adding ABA. A parallel experiment tested the effect of ABA alone (sampling at 0 h, 1 h, 2 h, and 3 h). During the entire experiment, cells were kept in the dark at 27 °C on a shaker at 150 rpm. After harvest, the cells were shock-frozen in liquid nitrogen for further analysis (probing tubulin pools by Western blot and measuring PLD activity).

To interfere with the function of PLD, we used *n*-butanol, which can bind the product of PLD, phosphatidic acids (PAs), thus intercepting PLD signalling [34]. Strains TUA3 and TTL+TUA3 were pretreated with 1% *n*-butanol for 1 h prior to adding 50 µM ABA and sampling at 1 h, 2 h, and 3 h after the addition of ABA. A parallel set of experiments followed the effect of *n*-butanol alone, sampling at 0 h, 1 h, 2 h, and 3 h.

To address whether the activation of a G-protein could mimic the effect of ABA, we treated the sample for 1 h with the Mastoparan analogue Mas 7 at 5 µM, comparing the response to a sample without Mastoparan. To test whether G-protein activity is needed for the response to ABA, we used the inhibitor pertussis toxin (PTX) at 10 μg/mL for 1 h, prior to the addition of 50 µM ABA and sampled at 1 h, 2 h, and 3 h after the addition of ABA. This experiment was conducted in strain TUA3 only.

### 4.5. Protein Extraction and Western Blot

The cell lines TUA3 and TTL+TUA3 (at day 7 after sub-cultivation) were harvested through a Büchner funnel via a short-time vacuum (10 s) at 20 °C. Soluble proteins were extracted according to the method [25], separated by SDS-PAGE, and probed by Western blotting using the monoclonal antibodies ATT, recognising tyrosinated α-tubulin (T9028, Sigma-Aldrich, Darmstadt, Germany), and DM1A, detecting detyrosinated α-tubulin (T9026, Sigma-Aldrich, Darmstadt, Germany). Prior to SDS-PAGE, the extracted proteins were denatured at 95 °C for 5 min. After centrifugation at 15,300× *g* for 5 min, the supernatant was loaded, adjusting the volume to give equal concentrations of total proteins per lane. To do so, the protein concentration was determined by a Bradford assay (50 mg Coomassie Brilliant Blue 250 in 25 mL of ethanol:phosphoric acid:water 1:17:3), which involved staining for 15 min and measuring A_595_ against a calibration curve constructed with Bovine Serum Albumine. The samples were run in triplicate. One set was stained with Coomassie Brilliant Blue staining to verify the equal loading of lanes, and the other two sets were used for Western blotting to detect detyrosinated or tyrosinated α-tubulin. For signal development, a polyclonal anti-mouse IgG coupled with alkaline phosphatase (Sigma-Aldrich, Germany) dissolved in TBS buffer (20 mM Tris-HCl, pH 7.6, and 150 mM NaCl) was used. The signals were quantified by measuring the integrated density as described in [37] of each band on the Western blot by ImageJ (https://imagej.net/). For quantification, the relative percentage of tyrosinated or detyrosinated α-tubulin was determined over the entire signal for both tubulin pools.

### 4.6. Monitoring and Quantifying Microtubule Responses

The response of the microtubules to the different treatments was followed in the tubulin marker strain TUA3 by spinning disc confocal microscopy [22] using an AxioObserver Z1 microscope (Zeiss, Jena, Germany) with a cooled digital CCD camera (AxioCamMRm; Zeiss), exciting the GFP signal with the 488 nm line of an Argon–Krypton laser (Zeiss, Jena, Germany). Images were collected into Z-stacks, and orthogonal projections were constructed with ZEN Blue software (Zeiss, Jena, Germany) (https://www.zeiss.com/microscopy/en/products/software/zeiss-zen-lite.html, accessed on 9 December 2024) and then exported as TIFF files for further analysis. Microtubule integrity was quantified by using a probing line with a width of 8 pixels to buffer against background noise and by collecting four intensity profiles in equal spacing parallel to the long axis of the cell. The principle of this quantification is to determine the steepness of the intensity peaks (reflecting microtubules) by determining the first derivative of the peaks. When microtubules disassemble, this is reflected by filling up the troughs between the peaks, such that the steepness progressively dissipates. Data represent three biological replications with 20 individual cells per replication.

### 4.7. Measuring Steady-State Transcript Levels

To investigate the gene expression, RNA was extracted from TUA3 cells at day 7 after sub-culturing using the innuPREP Plant RNA kit (Analytik Jena, Jena, Germany) according to the instructions of the manufacturer. RNA quality was checked through electrophoresis on a 1% agarose gel. The cDNA was synthesised using the M-MuLV cDNA Synthesis Kit (New England Biolabs, Ipswich, MA, USA) from 1 µg of RNA template, according to the instructions of the manufacturer. Real-time PCR was conducted (Bio-Rad CFX, Bio-Rad, Munich, Germany) according to the oligo nucleotide primers given in Appendix A and quantifying relative transcript levels based on the −ΔC_t_ method [40].

## 5. Conclusions

Through the analysis of the overexpression of TUA3-GFP in the tobacco BY-2 cell line in the presence or absence of ABA under cold conditions, unexpectedly, it was found that ABA transiently induced the cold stability of microtubules. N-butanol treatment alone, as well as the dual treatments with n-butanol and ABA in tubulin marker line NtTUA3-GFP BY-2 cells, has shown that the effect of ABA on detyrosinated α-tubulin was independent of the n-butanol pathway. Apart from these conclusions, the application of ABA in a double transgenic BY-2 cell line overexpressing NtTUA3-GFP/OsTTLL12-RFP allowed for the detection of the role of OsTTLL12, namely, that it was necessary for the activation of detyrosinated α-tubulin by ABA application. The distribution between detyrosinated α-tubulin and tyrosinated α-tubulin is dependent on the cycle of enzyme TTL and enzyme TTC.

## Figures and Tables

**Figure 1 ijms-26-00278-f001:**
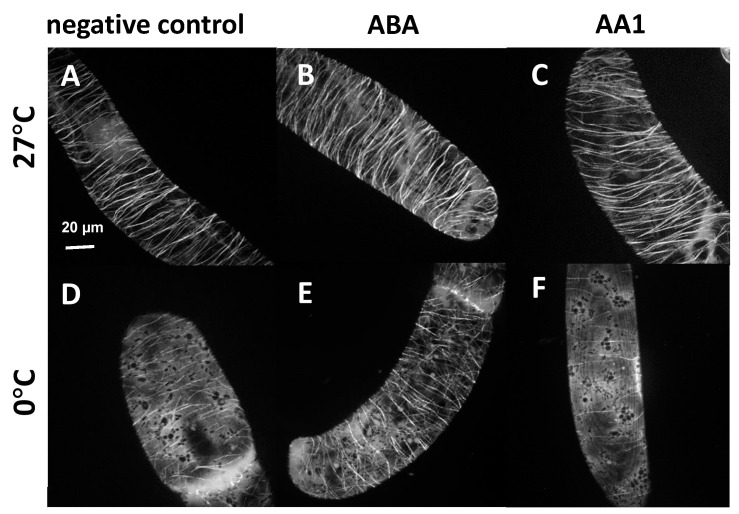
The response of cortical microtubules to modulation of ABA status and cold stress in cells expressing the *NtTUA3-GFP* marker. Representative cells either treated at 27 °C (**A**–**C**) or at 0 °C (**D**–**F**) either without treatment (**A**,**D**), after treatment with 50 µM of abscisic acid (**B**,**E**), or the abscisic acid antagonist AA1 (**C**,**F**) for 60 min.

**Figure 2 ijms-26-00278-f002:**
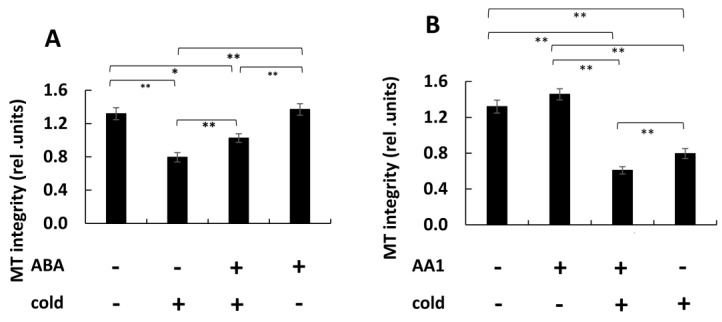
Quantification of the response of cortical microtubules to modulation of ABA status and cold stress in cells expressing the *NtTUA3-GFP* marker. Microtubule integrity was scored after 60 min of incubation either in the absence of cold stress at 27 °C (-) or at 0 °C (+) following a pre-treatment with either 50 µM ABA for 60 min (**A**) or with 500 µM of AA1, a blocker for ABA signalling for 60 min (**B**) as compared to the control in the aspect of the respective agent. Data represent mean and SE from three biological replications with a sample of 20 individual cells per replication. Differences were tested for significance with a Student’s *t*-test for paired data with * indicating significant differences at *p* < 0.05 and ** at *p* < 0.01.

**Figure 3 ijms-26-00278-f003:**
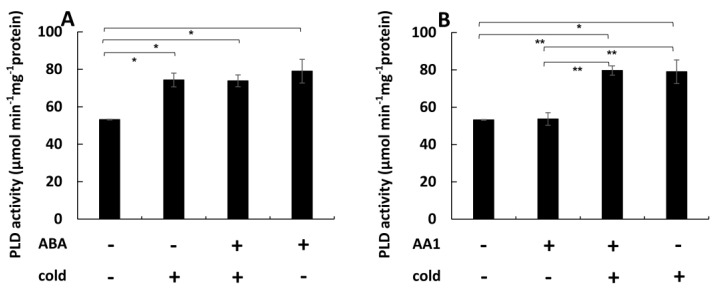
Response of Phospholipase D activity to modulation of ABA status and cold stress in cells expressing the *NtTUA3-GFP* marker. Specific activity was measured after 60 min of incubation either in the absence of cold stress at 27 °C (-) or at 0 °C (+) following a pre-treatment with either 50 µM ABA for 60 min (**A**) or with 500 µM of AA1, a blocker for ABA signalling for 60 min (**B**) as compared to the control in the aspect of the respective agent. Data represent mean and SE from three biological replications. Differences were tested for significance with a Student’s *t-*test for paired data with * indicating significant differences at *p* < 0.05 and ** at *p* < 0.01.

**Figure 4 ijms-26-00278-f004:**
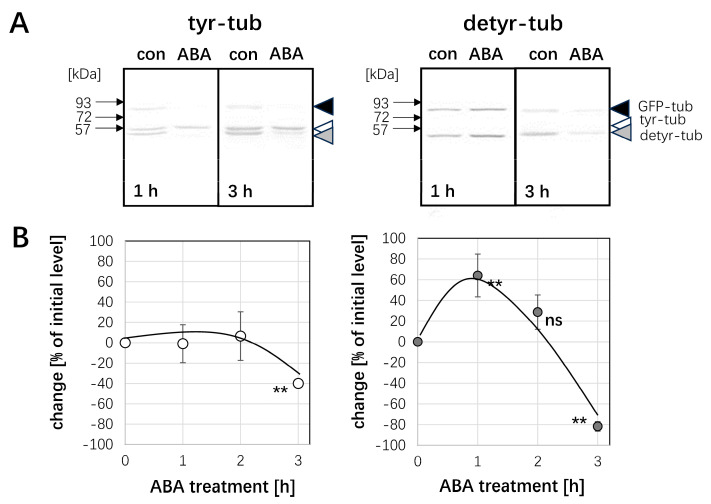
Response in the post-translational modification of α-tubulin to ABA in cells expressing the *NtTUA3-GFP* marker. (**A**) Representative Western blots after probing with monoclonal antibodies against tyrosinated α-tubulin (tyr-tub) and detyrosinated α-tubulin (detyr-tub) for 1 h or 3 h of treatment with 50 µM ABA as compared to control cells incubated without ABA (con). White arrowhead indicates the bona-fide tyrosinated form, the grey arrowhead the bona-fide detyrosinated form of a-tubulin, the black arrowhead the bona-fide GFP fusion of tobacco α3 tubulin. (**B**) Quantification of the relative change (in %) of the respective tubulin form in response to ABA as compared to the abundance seen in the control run in the adjacent lane of the blot. Data represent mean and standard errors from three independent extractions. Differences were tested for significance with a Student’s *t-*test for paired data with ** indicating significant differences at *p* < 0.01 and ns non significance with *p* > 0.05.

**Figure 5 ijms-26-00278-f005:**
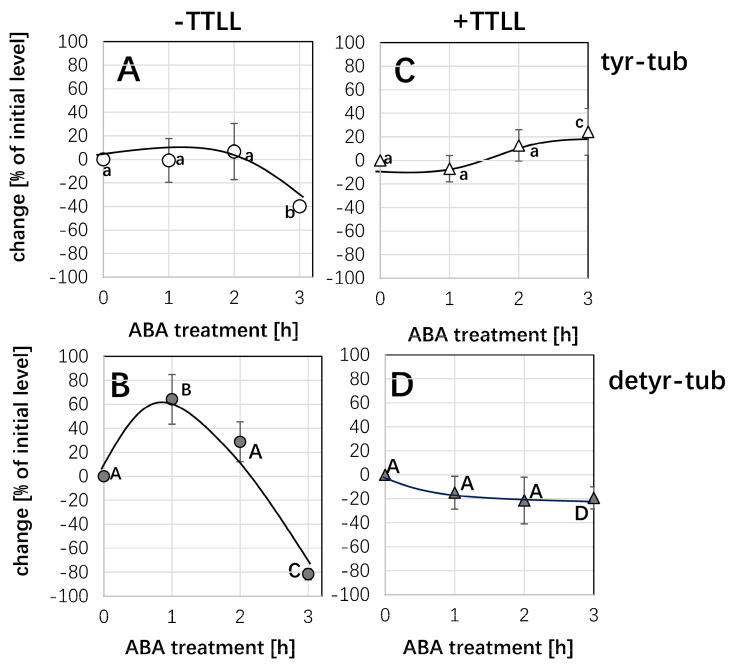
Response in the post-translational modification of α-tubulin to ABA in cells expressing the *NtTUA3-GFP* marker either in the absence (**A**,**B**) or presence (**C**,**D**) of rice TTLL. Quantification of the relative change (in %) of the respective tubulin form in response to ABA as compared to the abundance seen in the control run in the adjacent lane of the blot over time of treatment. Data represent mean and standard errors from three independent extractions. Statistical differences were tested by the Tukey LSD test, different letters indicate that the respective data point belongs to a different population at *p* < 0.05.

**Figure 6 ijms-26-00278-f006:**
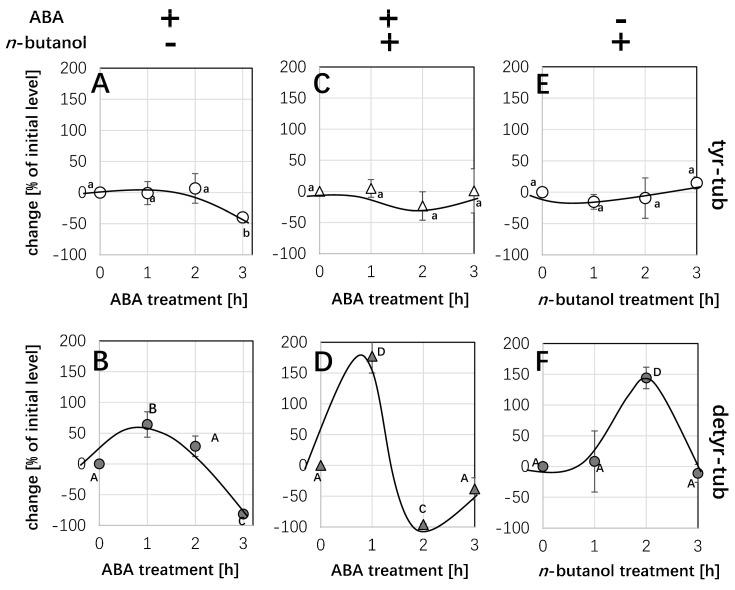
Interaction of ABA (50 µM) and *n*-butanol (1%) with respect to post-translational modification of α-tubulin in cells expressing the *NtTUA3-GFP* marker. Quantification of the relative change (in %) of the respective tubulin form as compared to the abundance seen in the control run in the adjacent lane of the blot over time of treatment. (**A**,**C**,**E**) tyrosinated α-tubulin, (**B**,**D**,**F**) detyrosinated α-tubulin. (**A**,**B**) response to ABA alone, (**C**,**D**) response to ABA after pre-treatment with *n*-butanol for 1 h, (**E**,**F**) response to *n*-butanol alone. Data represent mean and standard errors from three independent extractions. Statistical differences were tested by the Tukey LSD test, different letters indicate that the respective data point belongs to a different population at *p* < 0.05.

**Figure 7 ijms-26-00278-f007:**
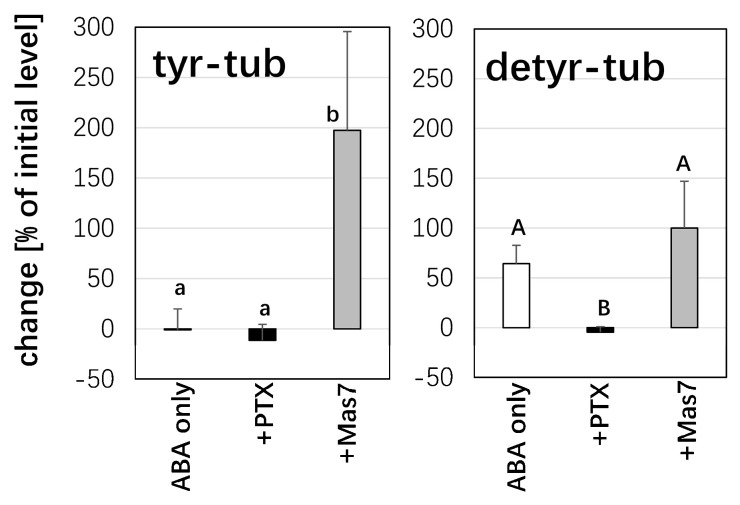
Effect of modulation of G-protein activity on the ABA-response of post-translational modification of α-tubulin in cells expressing the *NtTUA3-GFP* marker. The concentration of ABA was 50 µM, treatment duration was 1 h, either with ABA alone, or after pretreatment for 1 h with either the G-protein inhibitor Pertussis Toxin (PTX, 10 μg·mL^−1^), or the G-protein activator Mastoparan 7 (Mas7, 5 µM). Quantification of the relative change (in %) of the respective tubulin form as compared to the abundance seen in the control run in the adjacent lane of the blot over time of treatment. Data represent mean and standard errors from three independent extractions. Statistical differences were tested by the Tukey LSD test, different letters indicate that the respective data point belongs to a different population at *p* < 0.05.

**Figure 8 ijms-26-00278-f008:**
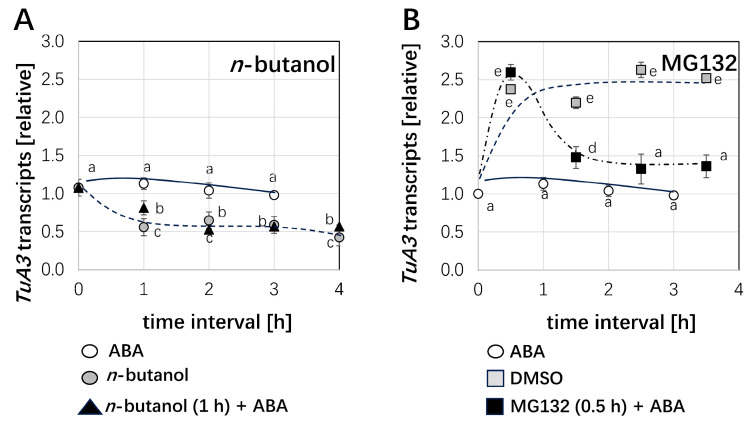
Dependence of the ABA response for steady-state transcript levels for tobacco *TuA3* on (**A**) Phospholipase D probed by preincubation with 1% *n*-butanol for 1 h, or on (**B**) proteasome activity probed by pre-incubation with 100 µM MG132 (using 1% DMSO as solvent) for 0.5 h. Relative levels, derived from the −ΔC_T_ values, are shown. Data represent mean and standard error from three biological replications, each in technical triplicates. Statistical differences were tested by the Tukey LSD test, different letters indicate that the respective data point belongs to a different population at *p* < 0.05.

## Data Availability

The original contributions presented in this study are included in the article. Further inquiries can be directed to the corresponding author.

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
