# Peer review of "Abscisic Acid, Microtubules and Phospholipase D-Solving a Cellular Bermuda Triangle"

_ijms, 2024, doi:10.3390/ijms26010278_

Round 1
Reviewer 1 Report
Comments and Suggestions for Authors
Dear authors,
This manuscript describes that abscisic acid, microtubules and phospholipase D interact to increase cold tolerance. This study represents several novel findings suggesting that (ABA) can transiently induce cold stability of microtubules, which were accompanied by the transient increase of phospholipase D (PLD) enzyme activity; ABA affected on detyrosinated α-tubulin by ABA and TTLL12 protein. Based on the results of ABA, 1% n-butanol, pertussis toxin (PTX), and mastoparan (Mas7) treatments, respectively, the authors document that the effect of ABA on detyrosinated α-tubulin was dependent on PLD activity. The manuscript is well-written. However, there are several inconsistencies that require the authors' attention.
1. Line 251-253. The conclusion is not consistent with Fig.3A, PLD activity of ABA treatment is higher than the other two treatments according to this Panel.
2. Line 269. Fig.4
3. Line 302-308. This sentence is a bit too long. Please had better revise the long sentence to short ones.
Author Response
Comments 1: Line 251-253. The conclusion is not consistent with Fig.3A, PLD activity of ABA treatment is higher than the other two treatments according to this Panel.
Response 1: Thank you for pointing this out. We agree with this comment. Therefore, we have change the sentence “Combination of ABA and cold stress did not yield an additive effect, the resulting stimulation of PLD activity was only slightly (and not significantly) higher than that seen for each factor alone.”to “ Combination of ABA and cold stress did not yield an additive effect, the resulting stimulation of PLD activitywas no difference with that seen for each factor alone.” You could find the change in Line 257-259 of revised manuscript.
Comments 2: Line 269. Fig.4
Response 2: Thank you for pointing this out. We agree with this comment. We have corrected this mistake. You could find the change in Line 275-277 of revised manuscript.
Comments 3: Line 302-308. This sentence is a bit too long. Please had better revise the long sentence to short ones.
Response 3: Thank you for pointing this out. We agree with this comment. We have revised the long sentence to short ones. You can find the change in Line 309-315 of revised manuscript.
Reviewer 2 Report
Comments and Suggestions for Authors
The paper is interesting, and the experiments are well designed and performed. But there are some minor points that should be addressed.
Title: Is more a journal style than a scientific style. The problem is that it suggests that the paper is a review rather than a research paper. I suggest using a more descriptive title, such as ABA transiently regulates microtubule stability under cold stress via...
The authors mention the n-butanol pathway. This pathway is not known to a wide audience, so please make an explanation of this pathway and contextualize its importance in the introduction.
Please correct the conclusion, as there are several typos:
LIne 497: duel--> dual.
Line 497: spacing after induced/cold
Line 498: spacing after BY-2
Author Response
Comments 1: Title: Is more a journal style than a scientific style. The problem is that it suggests that the paper is a review rather than a research paper. I suggest using a more descriptive title, such as ABA transiently regulates microtubule stability under cold stress via...
Response 1: Thank you for pointing this out. We agree with this comment. We have revised the title. You can find it in the Line 2-3 of the manuscript.
Comments 2: The authors mention the n-butanol pathway. This pathway is not known to a wide audience, so please make an explanation of this pathway and contextualize its importance in the introduction.
Response 2: Thank you for pointing this out. We agree with this comment. We have already explained n-butanol in the introduction. Please find them on Line 74-78.
Comments 3: Please correct the conclusion, as there are several typos:
LIne 497: duel--> dual.
Line 497: spacing after induced/cold
Line 498: spacing after BY-2
Response 3: Thank you for pointing this out. We have revised these mistakes. Please find them on Line 508-509.